# Effect of Dissolved Salts on Steady-State Heat Transfer Using Excessive Cooling by Water-Air Mists

Constantin Alberto Hernández-Bocanegra [1,2] , Francisco Andrés Acosta-González [3], José Ángel Ramos-Banderas [2] and Nancy Margarita López-Granados [2,*]

[1] CATEDRAS-CONACyT, Insurgentes Sur 1582 Av., Ciudad de Mexico 03940, Mexico; constantin.hb@morelia.tecnm.mx
[2] TecNM/I.T. Morelia, Tecnológico 1500 Av., Morelia 58120, Mexico; jose.rb@morelia.tecnm.mx
[3] Centro de Investigación y de Estudios Avanzados del Instituto Politécnico Nacional, P.O. Box 663, Saltillo 25000, Mexico; andres.acosta@cinvestav.edu.mx
* Correspondence: nancy.lg@morelia.tecnm.mx

**Abstract:** This work reports a new finding on the effect of dissolved salts, in water-air mists, on spray heat removal efficiencies from a metallic surface under steady state conditions. The experimental system is based on a calorimeter that measures heat flux removed by water-air mist sprays from 8 mm diameter × 2.5 mm thickness platinum samples heated by electromagnetic induction. During steady-state experiments, a solid-state controller equilibrates automatically the rate of heat generation with the rate of heat removal to reach a constant temperature. Equilibrium temperatures for stepwise T rising include 200 to 1200 °C in steps of 100 °C and then stepwise T that is lowered to 200 °C. The new finding is that, when using soft water-air mist and a high-water impingement density, a lack of temperature control during stepwise T increases was observed when stepping from 200 to 300 °C. This lack of temperature control is associated with a high heat flux and is attributed to the stabilization of the single-phase convection regime when T rising from 200 to 300 °C. Temperature stabilization was again possible only at wall temperatures $T_w \geq 600$ °C, at which single-phase convection was not stable. In contrast, when using a hard water-air mist under the same fluid flow conditions, all temperatures were readily reached. This is attributed to the transition from single-phase convection to nucleate boiling regime when T increased from 200 to 300 °C. This transition leads to a decrease in heat flux due to a reduction in the contact area between liquid and the wall surface. Finally, the corresponding boiling curves at high wall temperatures show the importance of heat radiation from the wall to understand the effect of salts during the stable vapor film regime.

**Keywords:** single phase convection; boiling convection; water-air mist sprays; salt water sprays



## 1. Introduction

The importance of spray cooling to remove heat from hot surfaces cannot be exaggerated. High surface temperatures (above ~600 °C) cause forced convection to increase the Leidenfrost temperature and, therefore, to improve the heat flux from the solid. Differently from water spray nozzles, water-air mist nozzles allow obtaining wider ranges of droplet velocity and size, combined with a wider range of water impingement density (kg·m$^{-2}$s$^{-1}$). This is a result of independent control of water and air flow rates and pressures. Heat transfer phenomena during water-air mist cooling may include all regimes, from single-phase convection, nucleate boiling, transition boiling, and stable vapor film regime above the Leidenfrost temperature. The so-called critical heat flux (CHF) represents the maximum heat flux removed from the wall and is reached at a critical temperature that separates nucleate boiling and transition boiling regimes. The use of extensive amounts of water in industrial spray cooling systems suggests some questions regarding the effect of water quality on the rate of heat removal. How can dissolved salt content affect heat flux removed

from hot surfaces? Are the critical temperature and CHF affected by dissolved salts? The same question arises for the temperatures of regime transitions.

The characterization of the rate of heat removal from hot walls is classified into two categories: transient and steady state experiments. In the former ones, an instrumented sample is preheated to a uniform temperature and then it is removed from the oven and placed in a quenching frame where it is impacted by a cooling jet, obtaining a cooling curve and temperature vs. time plot. The analysis of this curve is carried out by using the solution of the Inverse Heat Conduction Problem (IHCP). Therefore, heat flux from the wall is determined by an iterative procedure that minimizes the difference between the computed and measured cooling curves. In contrast, steady state experiments consist in equilibrating the sample temperature by supplying continuous power to the sample while simultaneously removing heat by the impacting jet. The removed heat flux from the wall is equivalent to the rate of heat supplied to the sample. The present work is an extension of previous studies using steady state experiments [1,2]. Nevertheless, for a wider scope of the phenomena, it is convenient to review previous works on both categories.

Cui et al. [3] conducted transient experiments cooling a copper cylinder from an initial temperature of 240 °C using water sprays with impingement densities of 0.5 and 3 kg·m$^{-2}$s$^{-1}$. Sauter mean diameter ($d_{32}$) and average velocity of the droplets were estimated to be ~200 microns and ~20 m·s$^{-1}$, respectively. The authors used water with dissolved salts, one at a time, NaCl, Na$_2$SO$_4$, and MgSO$_4$ at concentrations of 0.01, 0.06, 0.2, and 0.4 mol L$^{-1}$. It is interesting to note that the Leidenfrost temperature for most of these experiments was 220 °C, while single-phase convection was observed below 105 °C. The authors found that all dissolved salts enhanced the heat transfer rate in the nucleation and transition boiling regimes. This was attributed to the observed foaming in the liquid film generated during the nucleation boiling regime and to patches of deposited MgSO$_4$ during the transition boiling regime. They claim that the deposited salts increase the surface roughness, which improves the liquid–solid contact area and the number of vapor nucleation sites, increasing the rate of heat transfer during the nucleation boiling regime. Huang et al. [4] studied the effect of aqueous solutions of NaCl and KCl on the transition of the Leidenfrost point when 1.1 g of test liquid evaporated over an aluminum plate hold at constant temperature. Investigated aluminum temperatures were in the range of 200 to 300 °C. Liquid droplets were kept in position by machining shallow concave spherical surface depressions with 38 mm radius on the plates. Furthermore, the roughness of the metal surface was reported as 0.8 microns. Artificial nucleation cavities were also formed by machining cavity arrays, each having 0.5 mm in diameter and depth. The authors measured the evaporation time and found that at higher temperatures this time becomes essentially constant. This is a result of the presence of the stable vapor film regime. The Leidenfrost temperature corresponds to the threshold of this regime. The authors reported that Leidenfrost temperatures increased with salt content, which means an improvement in the rate of heat transfer as compared with distilled water. They suggest that the suppression of bubble coalescence by the dissolved salt is one of the mechanisms that affect the Leidenfrost transition. Moreover, they found that as salt concentration increases, the vapor film will quickly collapse and bubbles will rise to the surface. Guo et al. [5] analyzed the influence of dissolved MgSO$_4$, CaCl$_2$, Na$_2$SO$_4$, and NaCl in water on heat transfer from an aluminum plate heated to 495 °C and cooled by two water sprays. Droplet Sauter mean diameter was estimated as 319 microns and water impact density was in the range of 20.6 to 26.1 L·m$^{-2}$s$^{-1}$. The boiling curves were determined from the cooling curves using the solution of IHCP. They reported that, during nucleate boiling regimes, the results suggest a number of different phenomena. MgSO$_4$ solution increased heat flux compared with distilled water, but the NaCl solution led to little improvement. CaCl$_2$ caused the largest influence on the heat flux when it is at a concentration of 0.2 M, while Na$_2$SO$_4$ demonstrated this peak effect at a concentration of 0.06 M. These effects on boiling nucleation regime are attributed to surface tension gradients, transition concentration, and vapor pressure changes due to dissolved salts, which may lead to an inhibition of bubble

coalescence. Abdalrahman et al. [6] observed the influence of water salinity dissolving different concentrations of $MgSO_4$ in deionized water. A sample heated at an initial temperature of 560 °C was cooled continuously by means of water sprays with water impingement density of 3 $kg \cdot m^{-2} s^{-1}$. They found that using a mixture of salts has a greater effect than using any single salt. It is believed that the interaction between ions could change the properties (surface tension, saturation temperature, or liquid density) in solutions when used during cooling. Mohapatra et al. [7] reported results for transient experiments with a sample initial temperature of 1050 °C and using an air-mist with water impingement densities between 75 and 275 $L \cdot m^{-2} s^{-1}$. They discovered that adding salt to the water mist improves the surface heat flux in the nucleation boiling regime due to the suppression of bubble coalescence within an evaporating drop and also to foam formation, in agreement with reference [3]. In addition, the surface heat flux in the transition boiling regime is also improved, and this could be due to salt deposition on the hot surface during cooling. Lee et al. [8] investigated the effect of 35% sea salt solution on heat transfer from a reflood vertical long tube. A tube was heated up by applying a voltage to generate an electric current that heated up the tube by Joule effect. Initial temperatures were in the range of 620–720 °C. Electric power was turned off and the liquid was allowed to flow through the tube at an average velocity of 3 $cm \cdot s^{-1}$. They found that the cooling time was 10 s faster for the sea water solution compared with freshwater and the CHF (Critical Heat Flux) improvement for the sea salt solution was 9.7% compared to CHF with water during the immersion test. Pati et al. [9] used sprays of different types of refrigerants. Samples were heated up to an initial temperature of 950 °C and then cooled continuously using liquid impingement densities of 45, 58, 70, 80, and 96.8 $kg \cdot m^{-2} s^{-1}$. They found that cooling with seawater enhances the rate of heat transfer due to the phenomenon of deposition of salts on the hot surface and creating a larger area of heat transfer during evaporation of the droplets. Cheng et al. [10] reported steady-state experiments using a copper block hold at a temperature of 85 °C and using a spray with a flowrate of 4.5 $L \cdot h^{-1}$. They used High Alcohol Surfactant (HAS) and a Dissolved Salt Additive (DSA). The authors concluded that the effects on heat flows vary with concentrations and that the 2-ethylhexanol additive has the best performance of the four additives. The DSA has the inconvenience of clogging the nozzle and corroding the devices. Kumar et al. [11] carried out an investigation by using an analytical development of how the different mechanisms of heat transfer such as natural convection, enhanced latent heat, and forced convection influence nucleate pool boiling. The authors compared the results with experimental data from different fluids such as water, methanol, ethanol, and benzene. They concluded that a single mechanism cannot explain heat transfer in the entire nucleate boiling region. Moreover, they reported that, in the high heat flux region, almost all heat is transferred due to latent heat transport. Liu et al. [12] developed a steady-state experiment to study the effects of the concentration of surfactants such as Sodium Dodecyl Sulfate (SDS), n-octanol-distilled water, and Tween 20-distilled water on heat transfer using a micro swirl atomizing nozzle array. These nozzles operated at a liquid flow rate of 53 $L \cdot h^{-1}$ to equilibrate the sample temperature at values in the range from 200 to 300 °C. The surface was heated by using the thick film of electric resistors. The authors found that using SDS at a concentration of 200 ppm increases heat extraction capacities by 19.8%; however, both the addition of n-octanol and Tween 20 weakened the heat transfer performance of the spray cold plate. They also mentioned that the additives affect heat transfer by changing the surface tension of water. The reduction in surface tension can reduce the solid–liquid contact angle and can improve the spread ability and wettability of liquid droplets on the heating surface. Al-Gailani et al. [13] investigated the influence of heating and cooling rate and water composition on the kinetics of inorganic salt precipitation taking place when water is heated from ambient temperature up to its boiling point. They deposited a thin layer of liquid on a surface heated to 100 °C and the allowed it to cool down. They measured the rate of cooling of this wall. They found that the formation of inorganic deposits over the wall is not only expected in the heating period but also after cutting the heat source. Furthermore, the calcium carbonate formed

in the cooling period was greater than in the heating period. Misyura [14] performed a steady-state study on the heat transfer and evaporation rate of salt aqueous solutions such as LiBr, $CaCl_2$, LiCl, NaCl, CsCl, $BaCl_2$, and $MgCl_2$. He found that the heat transfer for water is only slightly greater than for salts, and for time >900 s, the water heat transfer coefficient becomes much higher than the one for saline solutions. He attributed this to the different heights of the liquid for water and salt solution, as well as to differences in liquid viscosity. Morozov and Elistratov [15] obtained experimental results in the regimen of nucleate boiling for aqueous solutions of LiBr and $CaCl_2$. They used a steady-state method by heating a horizontal plate to 130 °C. Then, the liquid was deposited on the horizontal wall and the rate of evaporation was measured. They found that the heat transfer coefficient for water is higher than for saline solutions by 15–30%. They claim that the suppression of the heat transfer for salts is associated with an increase in viscosity and surface tension with increasing salt concentrations in the solution.

This literature review shows that most of the reported studies on the effect of salts dissolved in water over the rate of heat removal from solid walls have been carried out under transient conditions, cooling the sample continuously from a broad range of initial temperatures and using water impingement densities over a wide range. All of them seem to agree that salts improve the rate of heat transfer as compared to pure water. In contrast, the few results reported in the literature using a steady-state method were carried out under low water impingement density values. Nevertheless, the respective conclusions on the effect of salts on heat transfer are not so conclusive. The present study was focused to gain a better understanding on the effect of dissolved salts in air-mists used for heat removal from solid walls in the temperature range from 200 to 1200 °C. The used experimental method is suitable to control sample temperature and cooling parameters, avoiding ambiguities in interpreting the measurements.

## 2. Materials and Methods

### 2.1. Heat Flux Experimental Method

Figure 1 shows the experimental system employed, which has already been explained in detail previously [1,16] and consists of the following: measuring probe, high-frequency generator, temperature controller, high-frequency ammeter, water and air supply lines, air-mist nozzles, positioning mechanism, and data acquisition system. The spray tests were used to determine the boiling curves, which are the plots of heat flux at the wall versus wall temperature between 200 and 1200 °C. Two independent water reservoirs were used: The first reservoir contained hard water from a well, and its average chemical composition for three samples is shown in Table 1. The second reservoir contained deionized water which was softened using an ion exchanger in order to reduce the content of calcium and magnesium sulfates in the water. For the experiments; firstly, the reference temperature of the automatic controller was set at 200 °C, while the sample was protected from the impingement of the water-air mist stream by means of an acrylic sheet. This allowed air and water flows to stabilize the target values while the sample reached the specified temperature in the controller. Once this was achieved, the acrylic was removed and the water-air mist impinged the sample, removing heat that was compensated when the controller demanded power from the generator to reach a sample temperature of 200 °C. To stabilize the power at the reference temperature, 480 s were required. Equilibrium temperatures were obtained in steps of 100 °C until reaching 1200 °C and back to 200 °C.

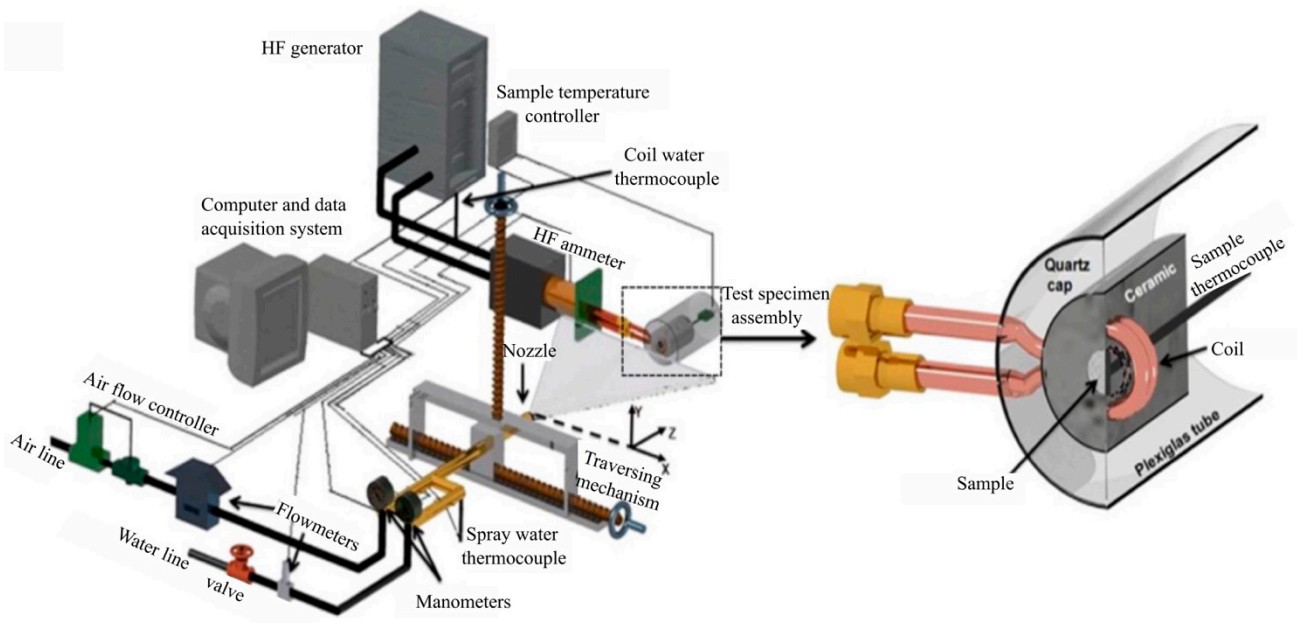

**Figure 1.** Scheme of experimental system and its components.

**Table 1.** Hardness analysis of the water used in air-mists.

| Sample | Hardness of Water CaCO$_3$ mg·L$^{-1}$ | Total Solids, mg·L$^{-1}$ | Suspended Solids, mg·L$^{-1}$ | Conductivity, μS·cm | Chlorides, mg·L$^{-1}$ | Sulfates, mg·L$^{-1}$ | pH |
|---|---|---|---|---|---|---|---|
| Hard water | 337.5 | 449.3 | 23 | 660.7 | 32.4 | 96.9 | 8 |
| Soft water | <10 | 28.2 | - | 44 | 6.9 | 5 | - |

*2.2. Spray Parameters Method*

Two different air-mist nozzles were tested in these experiments: Casterjet ½-6.5-90 (Spraying Systems Co., Glendale Heights, IL, USA.) and Delavan W19822 (Delavan Inc., Monroe, NC, USA.). Both nozzles are used in the secondary cooling system of continuous casting machines for steel. Schematics of the experimental setup used to measure the spray parameters are shown in Figure 2a,b. They consist, respectively, of a patternator for measuring water impingement density distributions and of a particle/droplet image analyzer (PDIA) system for acquiring and analyzing the images of fast-moving droplets in dense sprays; more details about these techniques were previously described [2].

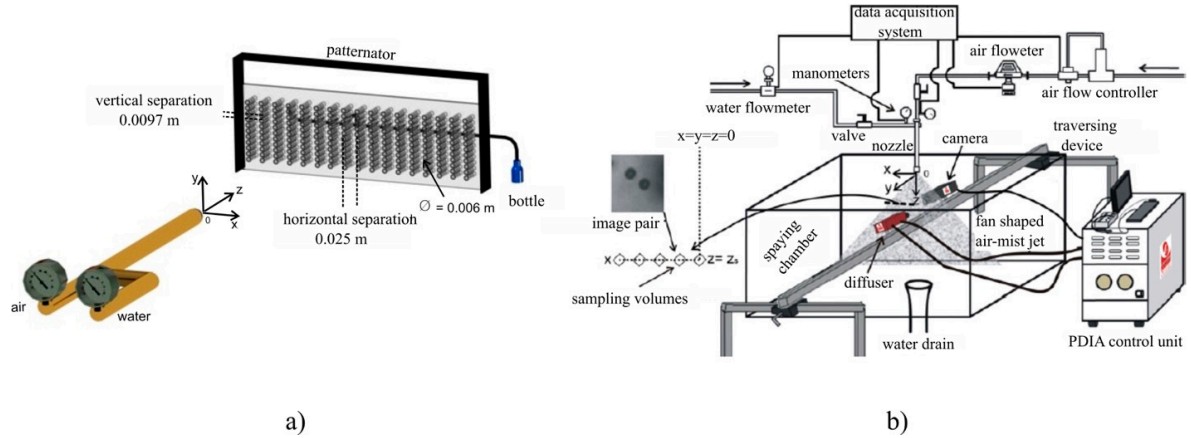

a)  b)

**Figure 2.** Schematics of experimental apparatuses: (**a**) patternator and (**b**) particle droplet image analyzer.

The local water impingement density was assessed by measuring the volume of water, $v$, collected over a prescribed period of time, $t$, in bottles connected to tubes with a cross-section area, $a$, placed at positions $x$–$y$–$z_s$, according to the following expression [16]:

$$w(x, y, z_s) = \frac{v}{t \cdot a \cos \theta} \tag{1}$$

where ($\cos \theta$) is the direction cosine of the angle formed between the nozzle axis and the line connecting the center of the nozzle orifice with the center of the entrance of a given collecting tube and was introduced to consider the projected area of the collecting tubes perpendicular to the direction of motion of the drops. The diameters of the collecting tubes and their separations are given in Figure 2a. Moreover, as seen in Figure 2b, the spray parameters were determined for non- impinging free water-air mists at different sampling locations along the major x-axis of the virtual impingement plane; the figure displays the coordinate system and the schematic of the sampling volumes. The water and air supply system shown in Figure 2b was essentially the same for the measurement of $q$, $d_d$, and $u$ and $w$, ensuring consistent flow rates for the different tests.

The volume mean diameter, $d_{3,0}$ (i.e., the diameter for which the volume times the number of droplets, $N$, is equal to the volume of the entire spray) was evaluated to consider the volume of the drops associated with the impinging water volume flux, according to the following equation:

$$d_{3,0} = \left( \sum_{i=1}^{N} d_{d,i}^3 / N \right)^{1/3} \tag{2}$$

where $d_{d,i}$ is the diameter of each drop $i$ in a sample of $N$ droplets. Additionally, the $z$ and $x$ volume-weighted mean velocity components defined, respectively, as follows.

$$u_{z,v} = \frac{\sum_{i=1}^{N} u_{z,v}\, d_{d,i}^3}{\sum_{i=1}^{N} d_{d,i}^3}; \quad u_{x,v} = \frac{\sum_{i=1}^{N} u_{x,v}\, d_{d,i}^3}{\sum_{i=1}^{N} d_{d,i}^3} \tag{3}$$

They were evaluated to consider the volume of each drop $i$ reaching the virtual impingement plane with normal and tangential velocity components $u_{z,i}$ and $u_{x,i}$, respectively.

## 3. Formulation of a Model for Impact Pressure

The studied nozzles were able to generate air-mist with a relatively high water impingement density and droplet velocity. These characteristics lead to impact pressures that may have an effect on the rate of heat removed from the wall. Total impact pressure is the sum of water and air impact pressures. In this work, expressions for these pressures were derived from the principle of conservation of mechanical energy, given by Bernoulli's equation, applied to the air-mist.

The impact pressure was estimated as a function of the water impingement density and the droplets velocity. Figure 3 shows a schematic representation of a water-air mist jet that impingements a vertical wall. Points 1 and 2 are located in the same streamline on the axis of the nozzle. Point 1 is far enough from the wall to avoid any upstream effect from the impingement point 2. Point 1 also corresponds to a position where water impingement density, mean droplet diameter, and velocity were measured in a previous work [17]. Bernoulli's equation between points 1 and 2 is given by the following expression:

$$\frac{P_2 - P_1}{\rho} + \frac{V_2^2}{2} - \frac{V_1^2}{2} + g(z_2 - z_1) = 0 \tag{4}$$

where $\rho$ is the water-air mist density, $g$ is gravity acceleration (9.8 m·s$^{-2}$), and $z_1 = z_2$. Moreover, at the impingement point, $V_2 = 0$. Since $P_1$ is practically equal to atmospheric

pressure, the manometric impact pressure at point 2 is the difference $P_2 - P_1$, and it is given by the following equation.

$$P_2 - P_1 = \frac{\rho V_1^2}{2} \tag{5}$$

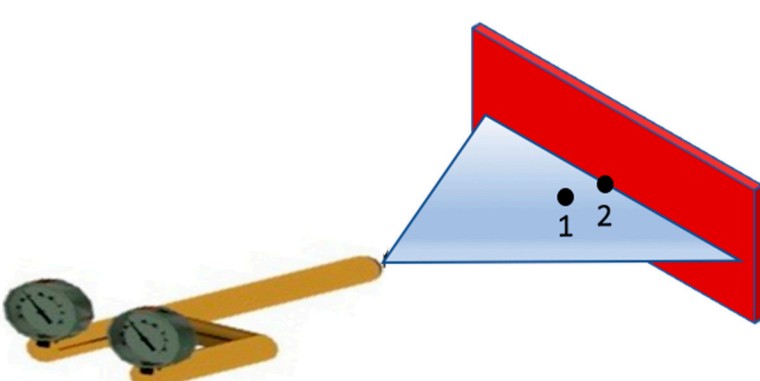

**Figure 3.** Schematic representation of a water-air mist impinging a vertical wall. Bernoulli's equation is applied in points 1 and 2 lying in the same streamline. Point 1 is far enough from the impingement plane, so there is no upstream influence from point 2.

The water-air mist density is a function of the number of droplets per unit volume and the droplets volume. If $f_L$ is the volume fraction of liquid in the water-air mist, it can be shown [18] that it is given by the following expression:

$$f_L = w/V \tag{6}$$

where $w$ is the water impingement density ($m^3 \cdot m^{-2} \cdot s^{-1}$), and $V$ is the droplet average velocity ($m \cdot s^{-1}$). Furthermore, the water-air mist density is obtained from averaging the densities of water and air according to the following equation.

$$\rho = \rho_L f_L + \rho_g(1 - f_L) \tag{7}$$

Subindexes $L$ and $g$ stand for liquid water and air, respectively. The substitution of Equation (6) into Equation (7) leads to the following equation.

$$\rho = \frac{\rho_L w}{V} + \rho_g\left(1 - \frac{w}{V}\right) \tag{8}$$

Therefore, the total manometric pressure at the wall is as follows.

$$P_2 - P_1 = \left[\rho_L w/V_1 + \rho_g(1 - w/V_1)\right]\frac{V_1^2}{2} \tag{9}$$

Equation (9) shows that the total manometric pressure is determined by the sum of two contributions: the water manometric pressure, given by the first term in the right hand side of the equation and the air manometric pressure. It is observed that the liquid fraction in the water-air mist is $w/V_1 \sim (0.020 \ m^3 \cdot m^{-2} \cdot s^{-1})/(20 \ m \cdot s^{-1}) = 0.001$. Since the water content in the jet is very low, pressure contributions from droplets and air momenta are both important to determine the total impact pressure. Notice that this is particularly valid for water-air mist jets, where droplet and air velocities are essentially the same. A different situation should be expected in water sprays, where air moves slower than droplets.

## 4. Results and Discussion

### 4.1. Effect of Dissolved Salts on Transition from Single Phase Convection to Nucleate Boiling Regime

Figure 4 shows the stepwise T rising curves for sample temperatures up to 400 °C (continuous lines) and the corresponding consumed electric power for electromagnetic heating (discontinuous lines). Soft and hard water-air mists were used to remove the sample heat, impacting the wall surface with a water flux of 20.6 kg·m$^{-2}$·s$^{-1}$. The curve from the experiment with hard water shows that all temperatures were readily reached. Electric power increases from every step to the next level and decreases asymptotically with time, while the sample temperature remains constant. In contrast, the experiment with soft water shows that stepping from 200 to 300 °C was not possible. Temperature and power oscillated abruptly, and the high frequency generator showed a loss of control and shutdown. Exceeding the limit of electric heating power suggests that a very high heat flux was demanded by the sample to increase its temperature at 300 °C. Sample surfaces were inspected after stopping the test at this point, and no salt layer was formed in neither sample. This means that sample temperature was not high enough to promote evident salt deposition on the sample surface from boiling water. Single phase convection should prevail at these wall temperatures.

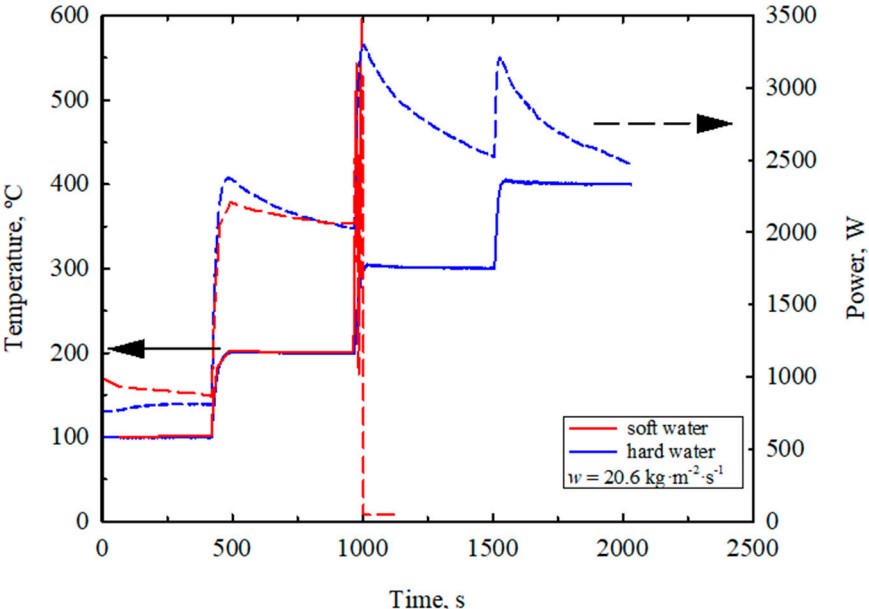

**Figure 4.** Measured sample temperature and consumed electric power during stepwise T rising using hard and soft water-air mists with water impingement density equal to 20.6 kg·m$^{-2}$·s$^{-1}$ delivered by a nozzle CasterJet 6.5-90.

Table 2 shows air-mist specifications that were used in the present work: Setback (distance between nozzle and wall surface), water flow rate, air pressure, water impingement density, volume-weighted mean droplet velocity, and volume mean droplet diameter were all measured in laboratory tests, as explained in Section 2.2. The rest of variables, such as water fraction in the air-mist, air density, manometric total impact pressure, manometric droplets impact pressure, and manometric air impact pressure, were computed from the equations derived in Section 3. The loss of control and shutdown obtained when using soft water was also observed for the nozzle operation conditions that are indicated in Table 2, with one exception: the air-mist from a Delavan nozzle that was operated with a water impingement density of 9.1 kg·m$^{-2}$·s$^{-1}$. The table shows that mean droplet diameter and velocity are similar to those measured for the critical cases. The main difference between critical cases and this isolated case is water impingement density. Table 2 shows the computed impact pressure from the droplets, from the air, and the total value. It can be seen that

computed total impact pressure is not an indicator to identify whether the soft water effect will appear or not. For example, consider the Delavan nozzle operating at W = 0.041 L·s⁻¹ and $p_A$ = 300 kPa and CasterJet nozzle operating at W = 0.1 L·s⁻¹ and $p_A$ = 98 kPa. The manometric total impact pressures are mutually similar at 554 and 547 Pa, respectively. However, the Delavan nozzle did not lead to power control loss when using soft water but the CasterJet did. The table also shows the computed water impact pressure. In this case, there is a clear trend on the effect of the liquid phase on loss of power control. The only test where this loss did not appear has the lowest value of water impact pressure: 130 Pa. This is a consequence of having the lowest water impact density: 9.1 kg·m⁻²·s⁻¹. It is also interesting to note that the estimated water fraction in the air-mist is also the lowest value: $3.17 \times 10^{-4}$.

The observed loss of control and shutdown, when using soft water air-mists, is attributed to a high rate of heat transfer. This rate surpasses the power limit of the high frequency generator, leading to temperature instability and a loss of power control. The shown results suggest that a higher water impact pressure improves the rate of heat removal from the wall, especially when using soft water. It is well established that the efficiency of heat removal from the wall is the highest when the wall surface is in full contact with liquid phase. Therefore, soft water should promote the stabilization of single phase convection at high impact densities. In contrast, when using hard water air-mists, the high water impact pressures that were studied in this work are not high enough to stabilize single phase convection, resulting in a nucleate boiling regime when wall temperature approaches 300 °C. In this regime, the bubbles that are attached to the wall act as a thermal resistance that decreases total heat flow from the wall. Under these conditions, the high frequency generator was able to supply enough power to increase the sample temperature from 200 to 300 °C. Further research is needed to clarify the mechanism by which the dissolved salts promote nucleate boiling regime. A number of factors, such as surface tension, vapor pressure, surface roughness, and wall properties, may have an important role in promoting nucleate boiling regime when salts are dissolved in the water.

### 4.2. Effect of Dissolved Salts on Boiling Curve for Lower Impact Pressure Water-Air Mists

The full boiling curve (from 200 to 1200 °C) was only determined for the case having the lowest impingement water density, which is the Delavan nozzle operated to deliver a water impingement density of 9.1 kg·m⁻²·s⁻¹ on the wall's surface. This condition allowed a stepwise T increase from 200 to 1200 °C, and then T was lowered to 200 °C, in steps of 100 °C, using either soft or hard water, with no loss of power control. Figure 5 shows the experimental boiling curve corresponding to this case. Hard water data are represented by black circles, while soft water results are indicated with black triangles. Moreover, the continuous lines represent stepwise T increases while discontinuous lines correspond to stepwise T decreases. In line with a previous work [1], stepwise T increasing heat fluxes are higher than those obtained for stepwise T decreases, leading to hysteresis. This was attributed to deposition of a salt layer on the wall surface from water evaporation. Unlike transient experiments where the sample is cooled continuously from an initial temperature in a short period of time, for example in seconds, in our steady-state experiments, the sample was exposed to the air-mist jet while it was held at constant temperatures for much longer times. This causes the accumulation of salt crystals on the sample surface by evaporation of water, forming a layer. Even for the case of using soft water, a thin salt layer was formed since the water-air mist impingement time during stepwise T increases was 1 h to reach 1200 °C. The lower heat flux obtained during stepwise T decreases was attributed to an increase in surface roughness due to the layer of salt. It was previously proposed [1] that air or vapor may be previously trapped within the gaps formed by the rough surface and eventually act as a thermal barrier for heat flow from the wall when T decreases. The validities of the results of the heat flux curves were previously reported [1].

**Table 2.** Effect of air-mist characteristics on loss of control and shutdown when using soft water.

| Nozzle | Spray Angle, $\alpha$, Degrees | Setback, $z_s$, m | Flow Rate, W, L·s⁻¹ | Air Pressure, pA, kPa | Impingement Density, $w$, kg·m⁻²s⁻¹ | Volumetric Mean Droplet Velocity, $u_{z,v}$ m·s⁻¹ | Volume Mean Diameter, $d_{30}$, μm | $\alpha d = w \cdot u_z^{-1}$ | Air Density, $\rho$, kg·m⁻³ | Air-Mist Density, $\rho$, kg·m⁻³ | Total Manometric Pressure, $P_2 - P_1$, Pa | Water Manometric Impact Pressure, $DP_{(H_2O)}$ Pa | Water Manometric impact Pressure, $DP_{(Air)}$ Pa | Loss of Control |
|---|---|---|---|---|---|---|---|---|---|---|---|---|---|---|
| Delavan W19822 | 90 | 0.19 | 0.041 | 300 | 9.1 | 28.7 | 30.91 | $3.17 \times 10^{-4}$ | 1.031 | 1.348 | 554 | 130 | 424 | No |
| | | | | 480 | 20.6 | 35.2 | 32.1 | $5.85 \times 10^{-4}$ | 1.031 | 1.616 | 1000 | 362 | 638 | Yes |
| | | | 0.076 | 412 | 22.4 | 31.4 | 34.2 | $7.13 \times 10^{-4}$ | 1.031 | 1.744 | 859 | 351 | 507 | Yes |
| | | | | 342 | 22.5 | 27.1 | 37 | $8.30 \times 10^{-4}$ | 1.031 | 1.860 | 683 | 304 | 378 | Yes |
| | | | | 189 | 21.1 | 22.3 | 51.3 | $9.46 \times 10^{-4}$ | 1.031 | 1.976 | 491 | 235 | 256 | Yes |
| CasterJet 1/2-6.5-90 | 90 | 0.175 | 0.1 | 98 | 20.63 | 24.1 | 62 | $8.56 \times 10^{-4}$ | 1.031 | 1.886 | 547 | 248 | 299 | Yes |
| | | | | 205 | 33.84 | 32.77 | 35.77 | $1.03 \times 10^{-3}$ | 1.031 | 2.063 | 1107 | 554 | 553 | Yes |
| | | | | 257 | 35.37 | 35.72 | 29.83 | $9.90 \times 10^{-4}$ | 1.031 | 2.020 | 1288 | 631 | 657 | Yes |
| | | | | 320 | 32.72 | 41.2 | 28.46 | $7.94 \times 10^{-4}$ | 1.031 | 1.824 | 1548 | 674 | 874 | Yes |

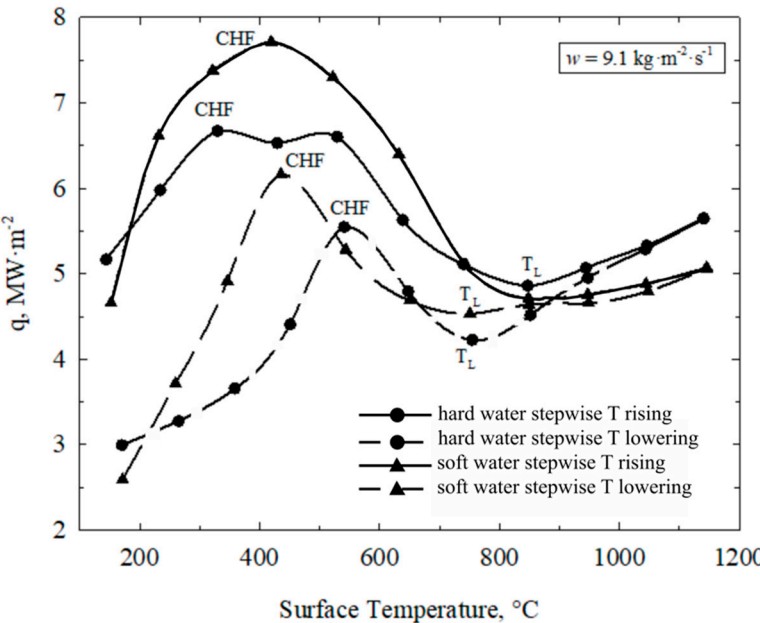

**Figure 5.** Experimentally determined steady-state boiling curves when using a low water impingement density of 9.1 kg·m$^{-2}$s$^{-1}$ delivered by a Delavan nozzle and using hard and soft water-air mists.

On the other hand, Figure 5 also shows a difference between the boiling curves obtained with soft and hard water. The heat fluxes obtained using soft water exceed those values measured using hard water-air mists, except at wall temperatures above 900 °C. These temperatures correspond to the stable vapor layer regime, above the Leidenfrost temperature. In this case, radiation heat losses become important. The emissivity of salt layer is ~0.9 [19] while the emissivity of platinum is ~0.2. This means that, for a given surface temperature, a salt layer improves the heat flux by radiation with respect to the value for a cleaner platinum surface. Below 900 °C, the heat flux obtained with hard water-air mist is smaller due to the effect of increasing surface roughness, as explained above. The overall peak CHF was obtained during stepwise T increases and using soft water-air mist, rating a value of ~7.8 MW·m$^{-2}$. The corresponding CHF obtained with hard water air-mist was ~6.7 MW·m$^{-2}$. Another difference in the heat flux curves obtained with soft and hard water-air mists is the surface temperature at CHF, which is higher when using hard water-air mist.

Figure 6a,b show some photomicrographs and chemical analysis of a sample surface after testing using hard water-air mist. Figure 6a shows Scanning Electron Microscopy (SEM) and Energy Dispersive Spectroscopy (EDS) (PHILIPS Model XL30, FEI COMPANY, Eindhoven, the Netherlands) analysis for the platinum surface (gray area) and Figure 6b shows the corresponding analysis for the surface of the salt layer. The increased surface roughness improves the rate of vapor nucleation, which decreases the contact area between liquid and solid wall, as discussed previously [1]. Moreover, it should be mentioned that hysteresis can be suppressed by cleaning the sample's surface to remove the salt layer after completing stepwise T increases up to 1200 °C [20].

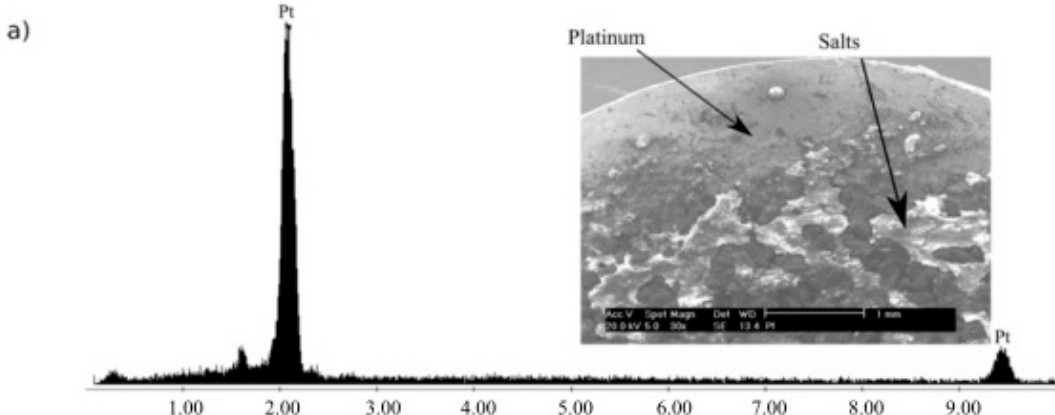

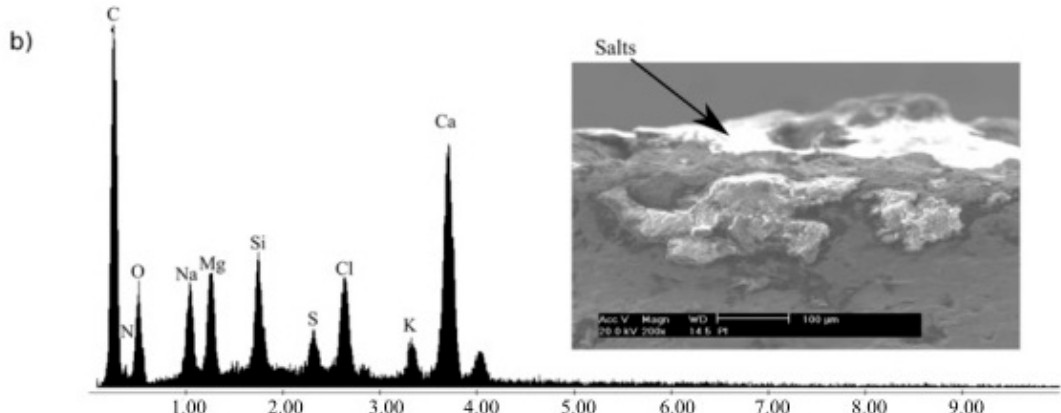

**Figure 6.** Photomicrographs and chemical analysis of a sample surface after testing. (**a**) EDS for the platinum (gray area) and (**b**) magnification of salt layer showing surface roughness and its EDS analysis.

### 4.3. Effect of Dissolved Salts on Boiling Curve for High Impingement Water Density Air Mists

As it was discussed in Section 4.1, applying air-mists with high impingement water density and using soft water led to a loss of control and shutdown when T increased from 200 to 300 °C. However, this control was recovered when sample was directly heated up to 600 °C, with no air-mist impacting on it, and then it was applied the soft water air-mist. It is clear that we "skipped" the temperature region where a loss of control occurs. In this manner, boiling curves were reported previously for these high water impingement conditions using soft water, but for sample temperatures from 600 to 1200 °C [1].

Figure 7 shows the obtained full boiling curve using hard water-air mist at a water impingement density of 21.8 kg·m$^{-2}$·s$^{-1}$. In contrast with Figure 5, the stepwise T increases in heat flux show a nearly horizontal curve. Heat flux is maintained within a narrow interval of 22 to 23 MW·m$^{-2}$ at wall temperatures from 300 to 1200 °C. The classic "inverted U" curve shape is barely drawn, and CHF and Leidenfrost temperature cannot be clearly defined. On the other hand, the heat flux curve obtained for stepwise T decreases follows the typical boiling curve shape clearly. In this case, CHF is 23 MW·m$^{-2}$ at 750 °C, and the Leidenfrost temperature is 900 °C. The behavior observed during stepwise T rising is attributed to the high impact pressure of the water-air mist on the wall. During this stage, the sample surface is relatively clean and a deposited salt layer is scarcely formed. The high impact pressure improves the contact surface area between water and metal; the latter agrees with that reported by Ramazani [21], who mentions that the absence of solid formation on the surface causes a significant improvement in the cooling rate. However, during stepwise T decreases, the accumulated salt layer creates nucleation sites

that improve the rate of vapor nucleation and, therefore, decrease the contact surface area between liquid and solid. "Flat" boiling curves for stepwise T increases have been previously reported for other water-air mist steady experiments under high impact pressure and using hard water [1].

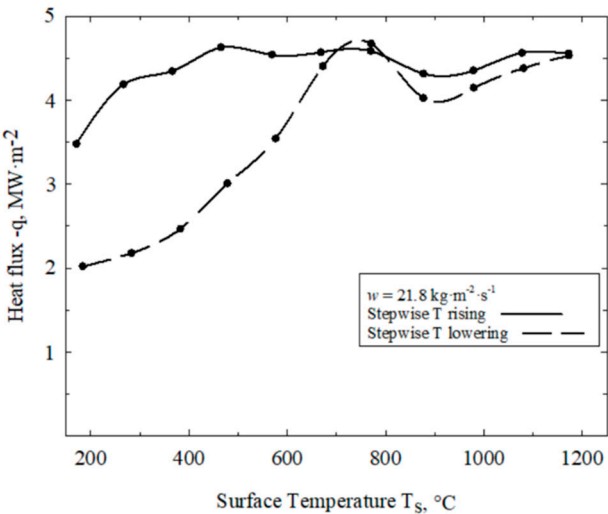

**Figure 7.** Experimentally determined steady-state boiling curve using hard water-air mist from a Delavan nozzle. Impingement water density, 21.8 kg·m²s⁻¹.

## 5. Conclusions

The effect of dissolved salts in water-air mists on the rate of heat transfer on high temperature metallic surfaces was studied using the steady state technique; from the above, the following is concluded:

- Dissolved salts promote the transition from single phase convection to nucleate boiling regime when T increases from 200 to 300 °C. This effect was inferred from the measured induction power required by a high frequency generator to reach all test temperature steps from 200 to 1200 °C. In contrast, when soft water was used in the air-mist, a loss of power control and shutdown was present when T increased from 200 to 300 °C. This reflects the high electric power required to control the sample's temperature, surpassing the limit capacity of the equipment. This is attributed to stabilization of the single-phase convection regime at 300 °C, which leads to a full intimate contact between water and the wall surface, and results in a higher heat flux removed from the wall.

- The referred losses of power control and shutdown were present in all experiments with soft water-air mists, except for the one with the lowest water impingement density of 9.1 L·m⁻²·s⁻¹. This low impingement density also led to the lowest value of manometric water impact pressure over the wall surface, 130 Pa. This pressure was computed from a developed two-phase fluid-dynamic model based on the principle of mechanical energy conservation, which is represented by Bernoulli´s equation.

Steady-state boiling curves were determined in the full temperature range from 200 to 1200 °C using soft and hard water-air mists but applying only the lowest water impingement density of 9.1 L·m⁻²·s⁻¹. In this case, three major findings are as follows:

- Hysteresis is present in both cases: soft and hard water. This is attributed to a deposition of a salt layer, which decreases heat flux when T decreased compared with when T increased. The gaps formed in the salt surface act as sites for vapor nucleation, decreasing the effective contact area between the liquid and the wall. This decreases heat flux when T decreased. Salt deposition is a consequence of water evaporation during long periods of contact between water-air mists and the hot surface.

- At temperatures below 900 °C, the heat flux removed by soft water-air mists is larger than the corresponding heat flux obtained by hard water-air mists. This may be attributed to an improvement in the rate of vapor nucleation when using hard water. The mechanism of vapor nucleation in the presence of dissolved salts in air-mists was not proposed in this document.
- At high wall temperatures above 900 °C, a stable vapor film regime was identified. In this case, radiation heat transfer plays an important role in determining the total heat flux from the wall. It was found that hard water-air mist led to a higher heat flux than the corresponding soft water-air mist. This is attributed to the larger value of the emissivity of the salt layer (0.9) compared with the lower emissivity of the platinum surface (0.2).

Steady state boiling curves for all the water impingement density conditions were obtained when using hard water-air mists. The corresponding main conclusions are as follows:

- Hysteresis is present, and the heat flux for T increases is higher than for T decreases. However, when using high water impingement density, the curve for T rising is nearly flat and does not show the classical inverted "U" shape for boiling curves. This is attributed to the high water impingement density, which seems to swap out most of vapor nuclei attached to the wall, improving direct contact between liquid and wall surface.
- When T decreases, the gaps formed on the salt layer promote vapor nucleation, leading to a lower heat flux.

From fluid dynamic analyses, it was found that the total impact pressure calculated from Bernoulli's equation does not offer a relationship between heat extraction and the calculated pressure. However, by breaking down this pressure into its components of water and air, there is a clear effect of water pressure on the rate of heat extraction.

**Author Contributions:** Conceptualization, F.A.A.-G. and C.A.H.-B.; investigation, C.A.H.-B.; project administration, C.A.H.-B. and F.A.A.-G.; resources, F.A.A.-G. writing—review and editing, C.A.H.-B., J.Á.R.-B. and N.M.L.-G. All authors have read and agreed to the published version of the manuscript.

**Funding:** This research received no external funding.

**Institutional Review Board Statement:** Not applicable.

**Informed Consent Statement:** Not applicable.

**Acknowledgments:** The authors want to acknowledge to TecNM-ITM, CATEDRAS-CONACyT, CINVESTAV Unidad-Saltillo, and SNI-CONACyT for the permanent support to the academic groups of Modeling of Metallurgical Processes.

**Conflicts of Interest:** The authors declare no conflict of interest.

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
