# Peer review of "Effect of Dissolved Salts on Steady-State Heat Transfer Using Excessive Cooling by Water-Air Mists"

_metals, doi:10.3390/met12050819_

Round 1

Reviewer 1 Report

Article:

 Effect of dissolved salts on steady-state heat transfer using excessive cooling by water-air mists and high surface temperatures  

Generally: Interesting paper suitable for Metals.

Comments:

P 2/79 -  They found that the cooling time was 10 seconds faster for the sea water – not clear faster than WHAT?

P3/110-114 – It seems that this text is not related to the paper.

Table 2 and text for example: gauge pressures of 555.06 Pa were obtained for the DELAVAN 222 nozzle and 547.74 Pa – pressure is computed value, it is strange to print the values with TWO decimal digits

Figure 4 – there is no reason for both 2D and 3D plot – select only one for the paper.

Figure 5 – MUST be corrected, labelling of the lines by dots and triangles and solid and dashed line is wrong. It is very confusing.

Figure 7, the point CHF at the flat curve should be removed (as is described in the text).

Conclusions- generally: It should be specify that the most of the conclusions is valid for surface temperature below the Leidenfrost temperature. Conclusions for above and below Leidenfrost should be described separately.

Reviewer 2 Report

This manuscript can be accepted for publication after major revisions, see the followings:

*The introduction should be improved (The literature review is a very weak).
*English should be improved.
*The Abstract and Conclusion should be improved. Outstanding results should be defined as quantified in the abstract and conclusions.
*The References should be updated.

*The novelty of this article is not clear. Please more explain it.
*Better description and explanation of figures 5 to 7. The description of the results of the tables is not sufficient and convincing.
*There are some typing errors and inaccuracies in the manuscript. Please, check the paper again for any possible misprints.

*The quality of figures should be improved.

* How has the validity of the results been examined? Please mention in the text of the full article.
* Introduction part needs to be extended by some of the recently published papers to show the importance of improve heat transfer in the good journals. The following references should be included in this manuscript:

[1] Kumar, Ashwini, Aruna Kumar Behura, Dipen Kumar Rajak, Ravinder Kumar, Mohammad H. Ahmadi, Mohsen Sharifpur, and Olusola Bamisile. "Performance of heat transfer mechanism in nucleate pool boiling-a relative approach of contribution to various heat transfer components." Case Studies in Thermal Engineering 24 (2021): 100827.

[2] Ramazani, Hamed Zahraei, Omid Mohammadi, Iman Rahgozar, Mohammad Behshad Shafii, and Mohammad Hossein Ahmadi. "Super-fast discharge of phase change materials by using an intermediate boiling fluid." International Communications in Heat and Mass Transfer 115 (2020): 104597.

*I hope that the authors refer to more published papers in Metal.

Round 2

Reviewer 2 Report

This article can be accepted.